# Research on Design of Stalk Furniture Based on the Concept and Application of Miryoku Engineering Theory

**Hong Li [1] and Kuo-Hsun Wen [2],***

1.   School of Creativity and Design, Guangzhou Huashang College, Guangzhou 511300, China; lihong576@gdhsc.edu.cn
2.   School of Arts, Macao Polytechnic Institute, Rua Luis Gonzaga Gomes, Macao 999078, China
*   Correspondence: khwen@ipm.edu.mo; Tel.: +853-88936902

**Abstract:** Through discussion on the disposal of crop stalks in rural China, where stalks are usually burned in the open air, resulting in the production of a large amount of smoke and air pollution, this study has been conducted to discuss the reasonable and effective use of waste stalks, and to explain the excellent characteristics of stalks from the perspective of green, ecological sustainable design and user perceptual needs, as well as the application of stalks in furniture design. In view of the accuracy of stalks in the design of furniture products, the paper has adopted the evaluation grid method in Miryoku engineering to guide the design practice. In the research process, interviews were first conducted on the basis of sample selection, then based on the expert opinion survey method, eight testers were selected for in-depth interviews. Key words of charm factors were extracted from 190 picture samples, and then a personal evaluation structure map and statistical data were built to construct an overall evaluation structure of the rural style stalk furniture. Next, the score value of the charm factor of stalk furniture was analyzed through the Likert scale questionnaire, the charm factor and specific performance characteristics of the rural style were sorted, followed by a proposed design method. Finally, the effectiveness of the design strategy was verified with the design of stalk furniture in rural style. The study emerged with the feasibility of the design of furniture products made of stalks. The results of the study have shown the correlation between the elements of furniture products and the perceptual image of users. Furthermore, the results have demonstrated the reuse value of waste stalks and improved the innovative design of stalks in the home furnishing field, which conveys the concept of sustainable development.

**Keywords:** stalks; furniture design; Miryoku engineering; perceptual analysis; sustainable concept



## 1. Introduction

Stalk is the stem left after mature threshing of crops, also known as crop fiber. There are two types of stalks: food crop stalks (wheat stalk, rice stalk, corn stalk, sorghum stalk, etc.) and cash crop stalks (cotton stalk, sugar cane stalk, hemp stalk, reed stalk, bean stalk, etc.) [1]. Being a kind of green natural material, direct burning of stalks will produce nitrogen oxides that pollute the air. In the autumn of each year, the smog in northern China is serious, for which one of the important reasons is burning stalks in the open air, thus it becomes a big challenge to dispose the abandoned stalks in rural areas. In the past, research on design mainly focused on the use of stalks in the field of construction, or the design of stalk furniture products, while the mental imagery brought by the furniture styles were less discussed, and the emotional needs of users could not be effectively discovered, leading to limitations on design practice. From the perspective of protecting the ecological environment, this study has adopted Miryoku engineering theory as guidance on stalk material design practice [2]. With the establishment of the corresponding relationship between the users' perceptual image of the product and stalk furniture, the waste stalks can be effectively reused as a design element, thus reducing the air pollution caused by burning stalks, and breaking a path of ecological regeneration for discarded stalks.

China is a traditional agricultural country with a long history and rich in stalk resources. According to the statistics of China's annual grain output, the annual crop stalks produced can reach about 700 million tons. In the context of industrialization, with the development of the rural economy and the increasing learning of farmers, the traditional use of stalks is changing. There are a large number of surplus stalks in some areas, and farmers generally use on-site incineration to dispose of the waste stalks (as shown in Figure 1). The large amount of smoke produced by burning stalks and dust in the air can easily form haze weather (as shown in Figure 2). In addition, the smoke generated by burning stalks also affects the normal operation of civil aviation, railways, and highways. This has become an increasingly prominent environmental pollution problem in China in recent years. Therefore, finding approaches to reduce the air pollution caused by stalk burning, and rationally and efficiently use this biological resource as well as help to comprehensively utilize waste stalks will have important social significance in terms of environmental suitability. At the theoretical level, the innovative Miryoku engineering method of application in the furniture design with stalk as raw material has been explored.

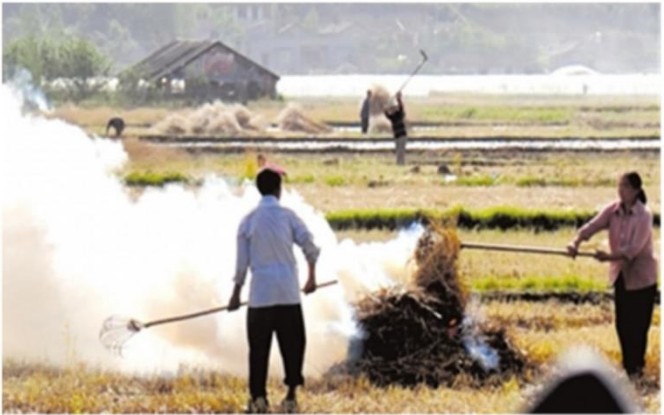

**Figure 1.** Buring stalks in rural area.

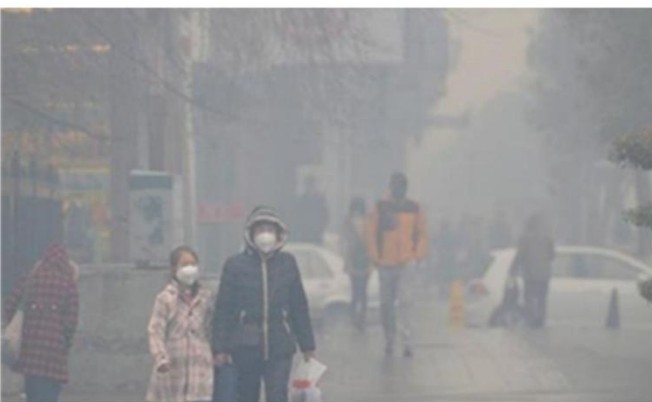

**Figure 2.** Haze weather in cities.

Currently, the development and utilization of biological resources has become the concern of countries all over the world, such as the rural energy projects in the United States, the ethanol energy plan in Brazil, the sunshine plan in Japan, and the green energy project planning in India. Among them, the United States uses stalk as feed and extracts ethanol from stalks. Denmark is the first country in the world to use stalk for power generation, and many developed countries have already made significant progress in the area of commercial application of stalks [3]. As a renewable resource, stalks can be reused in many ways such as feedstock, papermaking, artificial board, power generation, ethanol production, and other raw materials. As early as 1990s, there had been Nebraska-style

stalk building in North America, with waste stalks as alternative to the wood for a building material [4]. In Pakistan, stalk buildings have been developed to prevent earthquakes. This kind of building is durable, lightweight and energy-saving, and has a high seismic coefficient [5] (as shown in Figure 3). The Vanke City Pavilion at the 2010 World Expo in Shanghai, China was built with discarded wheat stalk (as shown in Figure 4), making the stalk building known as the protagonist of the design and conveying the concept of natural and environmental protection.

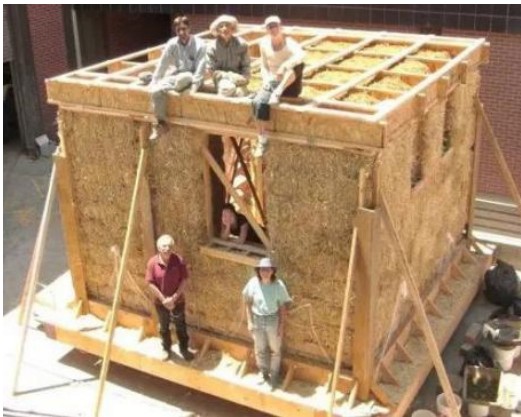

**Figure 3.** Stalk buildings in Pakistan (The Mater H, Internet source).

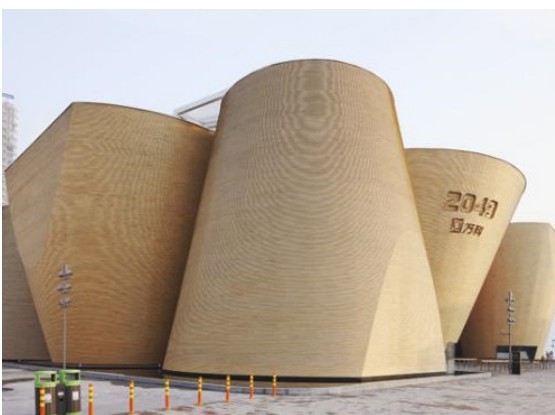

**Figure 4.** China Vanke City Pavilion (The Mater H, Internet source).

The main advantage of stalk buildings is that the waste stalks used are green materials, with excellent thermal insulation properties, and low energy consumption. They are used for house construction to avoid people disposing of them by burning and reducing the level of pollutants in the atmosphere. There are also stalk paintings, stalk weaving, and stalk crafts in China's intangible cultural heritage projects. Stalk materials have been used more widely in recent years with the spread of the concept of green and environmental protection. The stalks are made into raw materials through advanced processing technology and used in daily life such as toiletries, sanitary products, and disposable tableware. The wide application of stalk proves that it has broad prospects for the development in economy, environmental protection, and energy saving.

At present, the scope of applying stalk materials in the Chinese market is relatively small. The main reasons are: first, the labor cost of recycling waste stalk is too high; second, procedures such as crushing and injection molding in the secondary processing by the machine are complicated, leading to high technical requirements; third, consumers have limited awareness of stalk materials. Therefore, this research tries to find a solution based on the concept of simple and convenient use of waste stalks in the design of furniture products with an environmentally friendly and ecological way.

## 2. Materials and Methods

In terms of materials and manufacturing methods, this study discusses the material characteristics, drying process and molding process, in order to establish a complete knowledge basis for studying the reuse of straw materials.

### 2.1. Stalk Material

#### 2.1.1. The Composition Is Close to Wood

Whether crop stalks can be used in home and construction fields in replace of wood depends on its composition. The three major factors affecting the board's performance are the content of lignin, cellulose, and pentosane [6]. To understand whether stalks can be used in the field of furniture, its constituent elements must first be understood. Therefore, the study applies near-infrared reflectance spectroscopy (NIRS) to test the composition of crop stalks and trees. By comparing the composition of four commonly used crop stalks and three types of trees (as shown in Table 1), the data showed that the three index components of crop stalks are similar to the content of wood, with the index of cellulose content of sorghum stalks being especially close to that of wood. Therefore, crop stalks are theoretically suitable to replace wood boards for home furnishing and can meet the basic load-bearing strength and toughness.

**Table 1.** Comparative data of crop stalks and tree species composition.

| Raw Material | Lignin/% | Cellulose/% | Pentosane/% |
|---|---|---|---|
| Wheat stalk | 22.31 | 40.40 | 25.56 |
| Rice stalk | 14.05 | 36.20 | 18.06 |
| Sorghum stalk | 22.52 | 39.70 | 44.40 |
| Corn stalks | 18.38 | 37.68 | 21.58 |
| Birch | 23.91 | 53.43 | 25.90 |
| Poplar | 17.10 | 43.24 | 22.61 |
| Masson Pine | 28.42 | 51.86 | 8.54 |

#### 2.1.2. The Drying Process

To be used as household material, waste stalk material must be first dried and in common natural drying ways. Generally, it takes 30–50 days to dry and only can be used until the moisture content of the stalk is below 15%. However, this method is easily affected by the weather, thus it is not suitable for large-scale utilization [7]. The other method is the closed dehumidification and drying technology by adjusting the drying temperature of the machine to 75–85 °C [8], which is the optimum temperature for the drying of stalks. The entire process allows all parts of the stalks to be dried evenly, giving them strong stability, good outer surface quality, and enable surface hardening and thus not easy to crack. As a consequence, it can meet the requirements of moisture resistance and stability by household product materials.

#### 2.1.3. The Molding Processes

Stalks are rich in plant cellulose and have difficulty in meeting the basic molding process requirements. At present, the commonly used molding methods include flat pressing, molding, and extrusion [9]. The first is the flat molding method represented by the man-made stalk board. The stalk material is milled in accordance with the technical principles of physics and chemistry. This kind of stalk board has superior performance and can be applied to furniture panels without pollution after secondary processing on the surface. The second is to use a mold to press the stalks into shape, which is suitable for diversified product modeling and to meet different product requirements. The third is to make the product components that stalks can be subjected to in biological pretreatment, mechanical crushing, and the molding process and mold shaping. It then can create various standard and modeling parts of the product [10], which is suitable for 3D printing of raw materials with the process being simple, cheap, and suitable for mass production. In

addition, there are fourth method that uses cutting machines and grinding machines to carry out surface treatment to the leaves and stems of the stalks, using stems as structural elements of household products. By drawing upon their fiber strength, the stalks are threaded and arranged into strips as the component parts of the product.

*2.2. Advantages of Stalk as Furniture Material*

Based on the above description of straw materials and the discussion of reusing straw materials to reduce environmental pollution, the advantages of using straw materials will be discussed next. In general, the requirements for the furniture materials are as follows:

It should meet the basic functionality, which includes that the characteristics of the material can meet the basic usage requirements of people.

The process requirement primarily refers to whether the structure of the material is easy to be processed and produced.

The aesthetic requirement mainly refers to the overall style of furniture being in line with human aesthetic taste.

The economic requirement primarily refers to the price of control of materials.

The environmental protection requirement should consider the selection and usage, processing, and recycling of materials that are low-carbon oriented to be environmentally friendly.

With the continuous emergence of new materials and technologies, there are other aspects worth considering besides the above-mentioned basic aspects. Stalk is used as a new material in the home furnishing field and has the below advantages, from the perspective of sustainable development.

### 2.2.1. Abundant Raw Materials

As an agricultural country, China produces a large number of agricultural products annually; in the meantime, large amounts of crop stalks will be produced, most of which have not been effectively exploited. Stalk is the leftover stem part of crops after the fruit is harvested. It is an accessory product that does not require re-cutting of plants, and an annually renewable biological material, with the characteristics of short growth cycle, fast growth, and easy access, which is unlike traditional wood. The resources are very rich and not restricted by region, thus effectively reducing production costs.

### 2.2.2. Visual Aesthetics

The visual beauty of a material is the subjective feeling of the material that people perceive through specific behaviors; people judge the beauty and ugliness of different materials and obtain perceptual information to form a cognitive image of the material in the thinking space. The color of stalk material is light yellow or light brown, with different shades and rich luster. The tone of stalks gives us a warm feeling, with low color purity, and it is easy to coordinate with the real scene, without being too abrupt. Stalk material has a natural color, which forms a visual contrast while matching with gold metal, glass, plastic, and other materials, and could satisfy a consumer's individual pursuit of aesthetics in the current product homogeneity.

### 2.2.3. Green and Environmentally Friendly

Stalk is an accessory material after agricultural crops are discarded. It is taken from nature and can be degraded naturally, which has a relatively small burden on the environment and the local area. Modern furniture products have a short life cycle and a significant demand for raw materials; while stalk has strong reproducibility, a short material cycle, convenient disassembling and recycling, and is degradable and pollution-free in nature [11]. It is obvious that the advantages of stalks are consistent with the development needs of modern furniture and the concept advocated by sustainable design, which requires the harmonious development of people and the environment, and the implementation of the concept of sustainable design in the entire life cycle of product design. Consequently,

the use of product materials by considering its renewability and the characteristics of non-polluting raw materials should be fully apparent nowadays. In addition, parts and components of products in the scrapped stage that can be reused are crucial in achieving a sustainable cycle of the entire industrial chain. As a result, the use of stalks as furniture materials could provide a new "green" way to support environmental sustainability.

### 2.2.4. Spiritual Appeal

With the advancement of science and technology and processing techniques, the consumption of resources and the environmental pollution will inevitably be accelerated. "Environmental protection" has become the appeal of the times [12]; people's aesthetic standards have also changed including the pursuit of natural and irregular beauty and advocating for hand-made products. Undoubtedly, natural materials would re-enter people's lives and show their unique charm. In line with this approach, the natural properties of straw materials fit the concern for environmental protection. By applying waste stalks for furniture design, it can meet the spiritual needs of human beings, which are to get close to and love nature.

### *2.3. Research Process and Methods*

From the above analyses, it can be seen that stalk materials meet the basic conditions of furniture product design in terms of structural elements, molding process, visual aesthetics, and environmental protection [13]. Moreover, materials are an important part of furniture design, while the form of furniture is more and more homogenized nowadays due to functional constraints. It is undoubtedly an important method to meet the individual and diversified needs of current users in applying material sensibility as an innovation drive. It can provide a better way of ecological value for waste stalk materials and put forward new ideas for the innovative design of furniture products.

If waste stalk is used as a new material in furniture product design, it is necessary to consider the consumer's acceptance of it from their perspective. In order to avoid subjective and blind design by designers, it would be crucial to study the cognitive psychology of users, the preference relationship between users and product characteristics, and to rationally quantify perceptual cognition, as well as to extract emotional furniture design theories from it. The theory of Miryoku engineering has been adopted to identify the perceptual emotions of users to fully understand the needs of them, so as to help designers build user portraits more deeply, fully consider users' perceptual images of products, and then develop furniture products based on modern lifestyles and psychological needs. As a consequence, the success rate of development and design of stalk furniture will be improved.

Furniture design research mainly focuses on the design of product form, function, and structure, mostly adopting methods such as morphological analysis, Kano model, analytic hierarchy process (AHP), and TRIZ. The morphological analysis is based on systemic analysis and synthesis, with theories collected to decompose, arrange, and recombine the relevant morphological elements of the research object so as to obtain a new plan, and finally selection is carried out through evaluation. However, this method ignores the user's perceptual needs and opinions in the furniture design. The combination of the Kano model and perceptual engineering in the design of furniture products externalizes implicit user emotions to explicit product features. Many scholars have obtained certain research findings, but it is difficult to be applied in the design of irregular stalk materials. The analytic hierarchy process (AHP) can optimize different stalk materials by establishing a hierarchy and constructing a judgment matrix for each level, yet it is difficult to grasp the design of furniture styles. The TRIZ method proposes the core issues or fundamental contradictions that affect the resolution of the problem, and then uses the theories, methods, and principles to solve the problem step by step in accordance with the procedures. This method has strong characteristics of logicality, formality, and normativeness, while it is easy to ignore or exclude the subject, leading to the lack of personalization in the design of

stalk furniture. Therefore, based on the research of perceptual engineering, this paper takes sustainable design and user perceptual needs as the starting point, and uses the evaluation structure method to discuss the design of waste stalks.

The term "Miryoku engineering" is a concept proposed by Japanese scholars—Mr. Koizumi and Masao Inui in reference to "The Psychology of Personal Constructs" by clinical psychologist Kelly [14]. The content of Miryoku engineering research is to quantify human emotions that are difficult to analyze and capture where its conclusions could produce effective and quantitative support for design style analysis, material and non-material product design, and evaluation. Compared with perceptual engineering, Miryoku engineering is a design concept dominated by user preferences and pays more attention to "consumer preference". One of its main research methods is the application of the evaluation grid method. By allowing the respondents to compare perceptions of similar things during in-depth interviews and guiding the respondents to select differently in view of preferences, the evaluation grid method conducts in-depth analysis of the reason why the object attracts users and concretizes the vague concept to extract the effective charm factor of the object [15]. In general, the user's image of the product is disorderly stored in the brain in the form of data or information, which requires people to communicate with and stimulate each other to be externalized [16]. In the process of designing stalk furniture, the purpose is to find out the relationship between human perceptual knowledge and objects, to grasp human cognition or aesthetic taste through experiments and measurements, to quantitatively analyze the experimental results, and finally to seek the best principles as the guide of design practice.

In the case of furniture products becoming more and more homogeneous, people begin to stress more on the spiritual function of furniture products. The style of the product contains the spiritual meaning and social cultural connotation of the product and has become the main carrier of the spiritual function of the product [17]. As an important aspect of users' emotional cognition, the style orientation of products expresses a subjective and individual behavior. Moreover, furniture style includes product functions and basic elements, and is an important manifestation of perceptual cognition. Therefore, it is reasonable to study from the perspective of design style of furniture products. First, it requires establishing a framework for innovative design of stalk furniture based on Miryoku engineering (as shown in Figure 5).

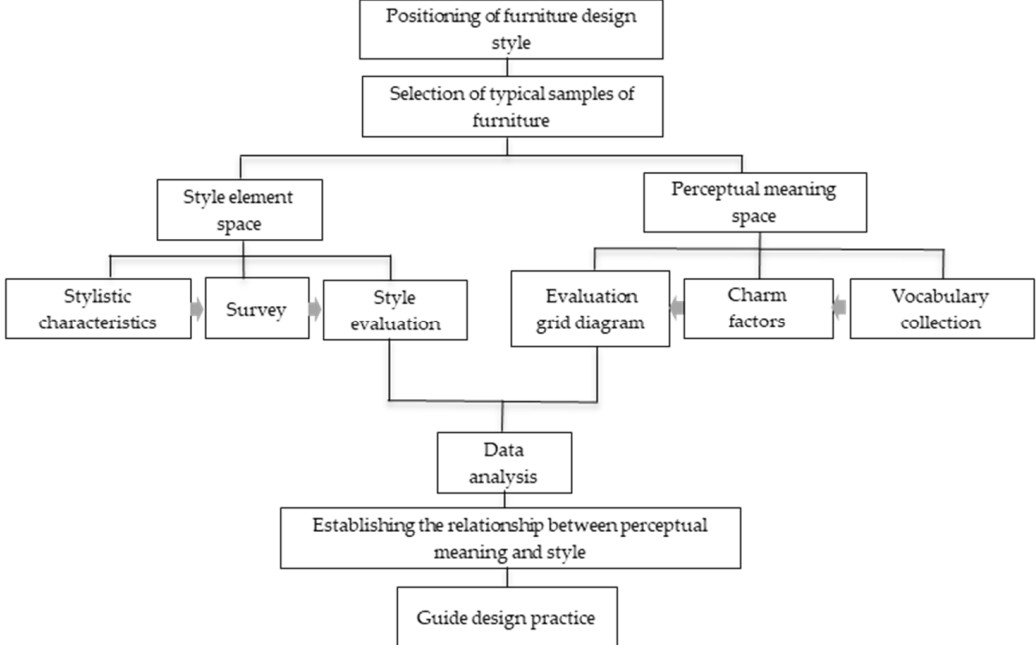

**Figure 5.** Innovative design framework for stalk furniture based on Miryoku engineering.

The rich expressiveness and feasibility of stalk materials in furniture design can be verified through the theory of Miryoku engineering. The positioning of the furniture style is rural style designed with stalk as the material. First, it is important to adopt the evaluation grid method in Miryoku engineering to study the charm factor of rural style furniture, and to establish the evaluation grid diagram of the style. Then, it follows with the application of using the upper charm factor of the pastoral simplicity style, that is, the user's abstract perception scale of the style, as the evaluation item [18]. Finally, the design can be verified according to the evaluation result of users' experience, and the research flow chart shown in Table 2.

**Table 2.** The experiment of adopting the evaluation grid method in Miryoku engineering.

| Preliminary Preparation | The First Stage | The Second Stage | The Third Stage |
| --- | --- | --- | --- |
| Define and collect rural-style furniture picture samples | Extract charm factors through in-depth interviews and establish a personal evaluation grid diagram | Evaluation scale and questionnaire design | Propose a design strategy |
| Sample screening | Simplify the charm factors with KJ method | Questionnaire release and retrieve | Verify the design |
| Screening of interviewees | Establish an overall evaluation grid diagram | Questionnaire analysis | Experimental results |

2.3.1. Establish Experimental Samples and Screen Interview Subjects

(1) Defining sample: Before collecting pictures of rural-style furniture, the stalk furniture should be defined. From the perspective of materials, structure, function, and form, the stalk furniture described in the study is defined as a device using stalks as the base material and can meet the needs of users in the home environment.

(2) Samples collection: The study collected 60 common stalk pictures via online search engine such as Baidu with the keyword "stalk material", 45 picture samples of "furniture form", and 85 picture samples of "rural-style home furniture", gathering a total sample of 190 copies. Through the card classification method (KJ technique), similar samples were grouped together, and processed into graphic cards in a uniform size [19].

(3) Interviewees: As the experimental samples are relatively professional, this study targeted users with design and industry experience, and finally selected 3 environmental art designers, 2 furniture sales consultants, and three students majoring in furniture design as the subject of in-depth interviews. Those with professional experience are suited to accurately distinguish and express sample differences and provide extraction effects of charm factors.

According to the market research conducted by Nielsen [20], in the field of human–computer interaction, 76% of usability problems can be found by inviting 5 skilled users to test each product. In order to improve the reliability of survey interviews, 8 respondents were selected.

The research was conducted by using the expert opinion survey method, which, also known as the "Telfi method", refers to the way in which a certain topic or question solicits opinions from relevant experts or authorities. Relying on the knowledge and experience of experts has an important decision-making effect on the judgment, evaluation, and prediction of research problems. This research screened environmental art designers to represent furniture design experts, home sales consultants to represent sales experts in furniture market, and students majoring in furniture design to represent designers as well as future furniture consumers. Because the research has been carried out mainly from a design perspective, three environmental art designers and three students majoring in furniture design were selected; while the home salesperson represents the consumer market, only two are selected for appropriate weakening. These three groups are of representative

significance in the fields of furniture design, sales, and consumption. Selecting them as the research objects can improve the accuracy and reliability of the research.

(4) In-depth interviews: One-to-one interviews were conducted for each subject, taking notes and recording during the whole process. First, the interviewees chose their favorites among the 190 picture samples. Second, they classified the selected pictures, and compared the classified samples with explanation for the original reason of classification, which constituted the median charm factor of the evaluation grid diagram [21]. Finally, according to the original evaluation, the step method was used to further ask the interviewees' abstract feelings (upper factor) and specific characteristics (lower factor). For example, when asking the interviewee for abstract reasons (superior factor) such as "How do you feel about this type of chair?", if the respondent replies, "I think it is simple and elegant", the study will extract "simple and elegant" as an abstract feeling. When the conducted interview continues to ask the interviewee for specific reasons (lower factors) such as "Which aspect of the simplicity of this chair do you think you like?", if the respondent answers, "I like its smooth lines", then "smooth lines" will be used as a specific reason.

### 2.3.2. Establish an Evaluation Grid Diagram (EGM)

In light of on-site notes and audio recordings, the recording results of 8 interviewees were sorted, and keywords were extracted from their original evaluation (median factor), abstract feeling (upper factor) and specific characteristics (lower factor) [22], and then the personal evaluation grid diagram of 8 interviewees was set up. At the same time, the individual evaluation grid of each respondent was analyzed, and the upper, median, and lower factors of the 8 evaluation grids were integrated, classified, and simplified by using the KJ method (affinity diagram). The specific steps are as follows: the first step involves writing the charm factors provided by the 8 interviewees on small cards; the second step involves classifying all abstract, original, and specific reasons and the number of mentions; the third step involves screening, classify, simplify, and compare the quotations of each interviewee.

At first, an evaluation grid diagram of stalk materials was created (as shown in Figure 6). From the statistics of the lower factors of the material evaluation grid diagram, the top 4 selections are rice, wheat, sorghum, and corn stalks; the second step involved making an evaluation grid diagram for the shape of stalk furniture (as shown in Figure 7). The top 4 selections are rhombus, cylinder, square, and ring; the third step is to establish the overall evaluation grid diagram of rural style stalk furniture (as shown in Figure 8).

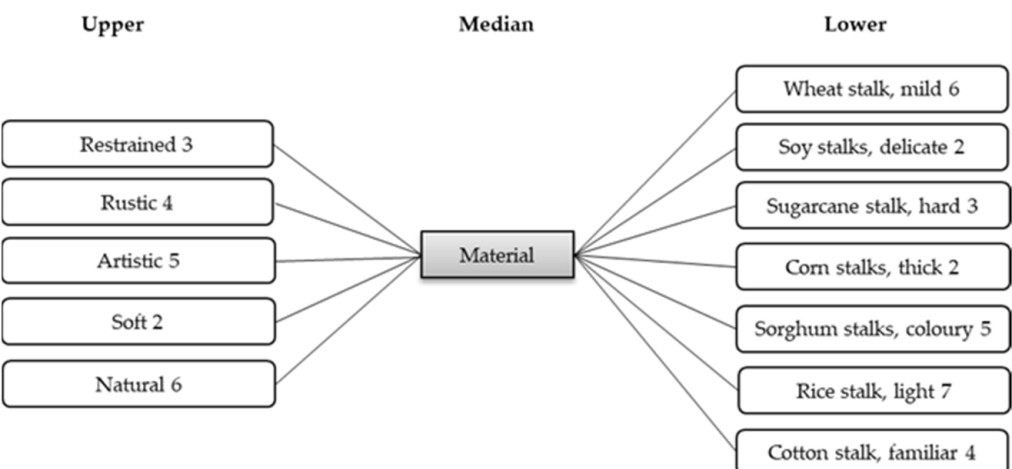

**Figure 6.** Evaluation grid diagram of stalk materials (number refers to frequency of mentions).

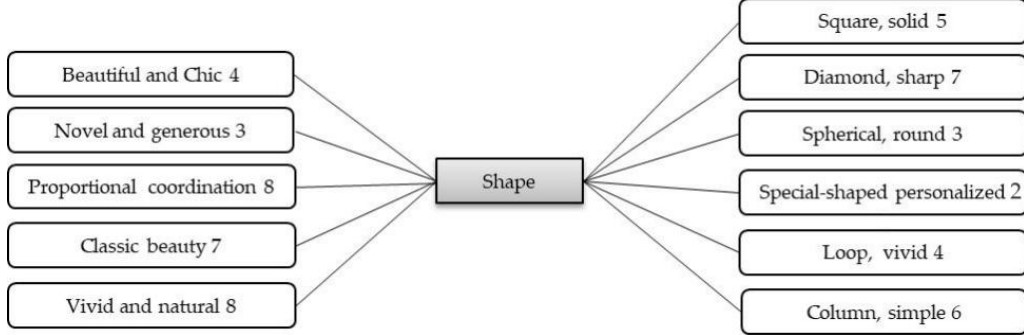

**Figure 7.** Evaluation grid diagram of the shape of stalk furniture.

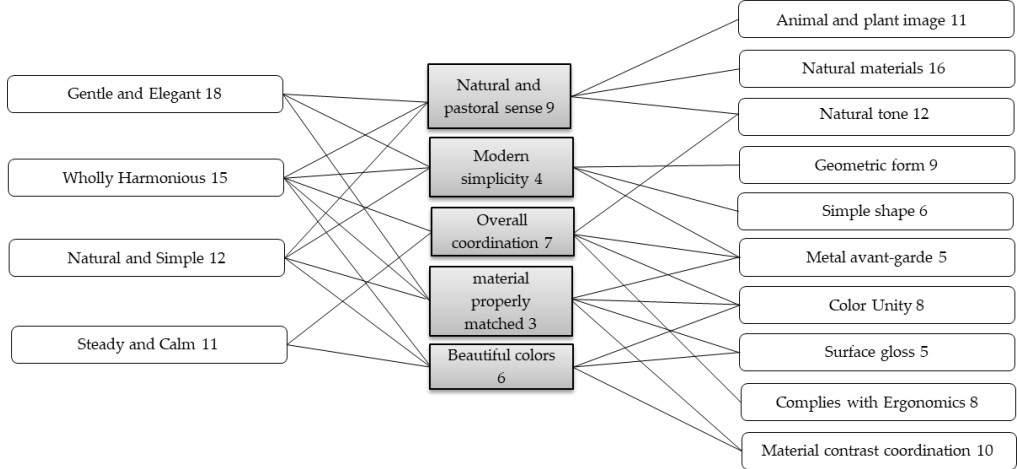

**Figure 8.** Overall evaluation grid diagram of rural-style stalk furniture.

### 2.4. Evaluation of the Rural Style Design of Stalk Furniture

First, the fit between the material and the shape of the stalk was determined. From the above-mentioned material and shape evaluation grid diagram, the top 4 were selected to be combined correspondingly, with the rice stalk and the rhombus as sample 1, the wheat stalk and the cylinder as sample 2, the sorghum stalk and square as sample 3, and the corn stalk and ring as sample 4, (as shown in Figure 9). The first step judged the fit between the corresponding materials and shapes of the four groups of samples by the interviewees, and the second step measured the scores of the participants on the charm factors of the matching pastoral style through the evaluation scale. From the overall evaluation grid diagram of the rural style stalk furniture, it can be concluded that the main four charm factors are "gentle and elegant, wholly harmonious, natural and simple, calm and steady". Furthermore, these four samples were scored from the angle of four main charm factors, from the highest sample to determine one of the most recognized sample combinations.

The above four samples were scored on the basis of quantitative analysis of the charm factors with the attitude scale. The questionnaire answers were scored by Likert Scale, with the respondent's own feelings towards the strength and weakness of a sample as criteria, 5 points for the strongest and 1 point for the weakest. Through the online scoring platform Sojump, a total of 113 questionnaires were retrieved, 27 invalid questionnaires were eliminated, and the remaining 86 questionnaire results were counted, establishing the average and overall average values of the 4 samples in each rural style charm factor (shown as Table 3).

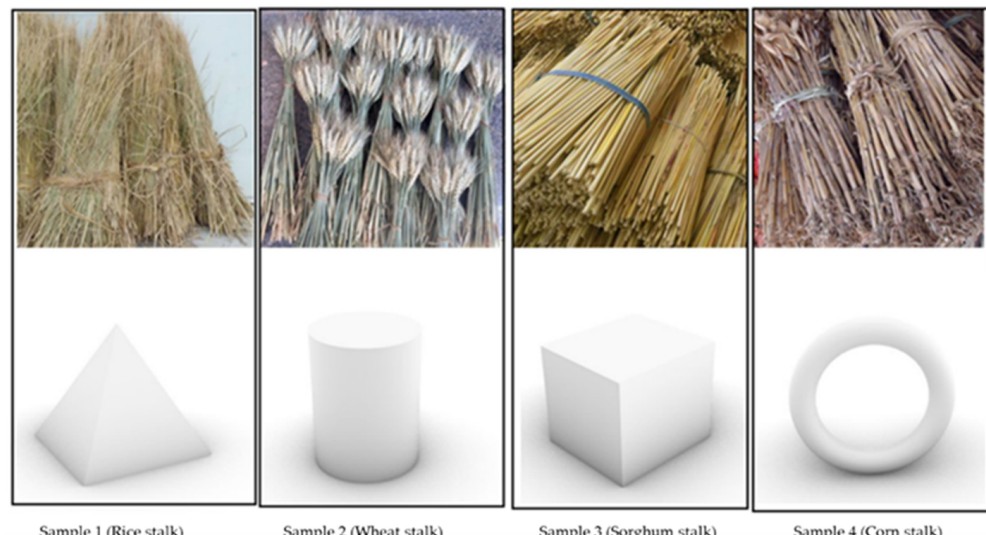

**Figure 9.** Combination diagram of stalk materials and geometric shapes (author photos and virtue design).

**Table 3.** Questionnaire statistics of the charm factor of rural style stalk furniture.

| Charm Factor | Sample 1 | Sample 2 | Sample 3 | Sample 4 |
|---|---|---|---|---|
| Gentle and Elegant | 2.82 | 3.26 | 2.91 | 3.12 |
| Wholly Harmonious | 3.27 | 2.96 | 3.46 | 2.95 |
| Natural and Simple | 3.43 | 3.14 | 3.32 | 3.06 |
| Calm and Steady | 2.95 | 2.89 | 3.41 | 2.25 |
| Average | 3.12 | 3.06 | 3.28 | 2.85 |

It can be seen directly from the table that the charm factors of "natural and simple" and "wholly harmonious" are highly accepted. From the results of the questionnaire, it can be seen that the highest overall average value of the sample is sample 3, which means that the sample material has the highest degree of fit with the shape, indicating that the matching design of the two fits well and the overall pastoral style recognition is high.

## 3. Results

Referring to the overall evaluation grid of the rural style stalk furniture, charm factors and the specific performance features of the rural style were organized; meanwhile, the design elements were extracted. The following design strategies are then proposed:

(1) Material selection. Through attitude scaling analysis of four samples, including rice stalk, wheat stalk, sorghum stalk, and corn stalk, it is proven that the sample 3 sorghum stalk combination received highest mean scores. The inside of sorghum stalks is composed of elastic honeycombs [23] and the outer cortex of the stems is composed of thick-walled cells, which are keratinized, silicified, and bright in color. The uniform color meets the visual effects and mechanical support of furniture products.

(2) The combination of materials. The combination of sorghum stalks in sample 3 and a frame, bundled with several single plants, which is conducive to the conversion from a simple line to a specific shape, increases the structural reliability and strength of the product. The metal is made of flat iron, the shape is made by ion-cutting, and the metal frames are joined together by welding. The metal material and the sorghum stalks were combined by inlays, which allow several individual plants to be embedded in the metal frame in a scattered form or embedded as a whole in a bundled form (as shown in Figure 10). It is a form of combining individual plants side by side through a thread (as shown in Figure 11), used as surface material, embedded in a metal frame, strong and sturdy, easy to disassemble and recycle.

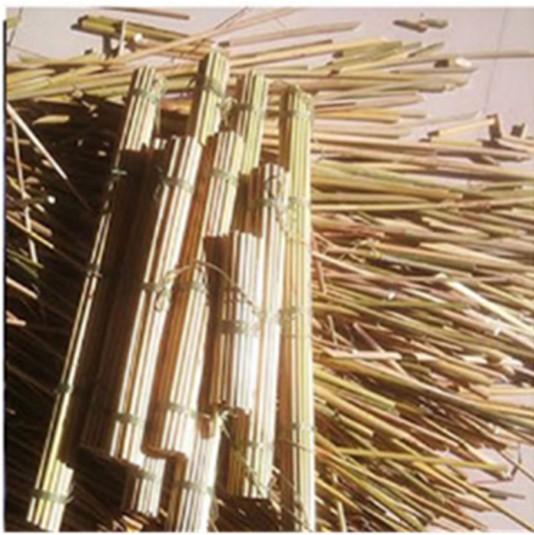

Sorghum stalks & cutting

**Figure 10.** The metal material and the sorghum stalks are combined by inlays (author photo).

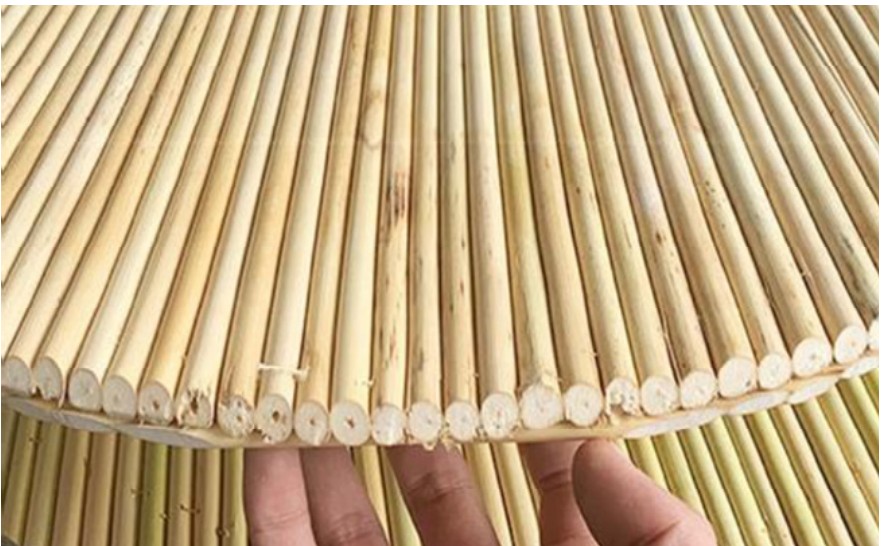

**Figure 11.** The form of combining individual plants side by side through a thread (author photo).

(3) Shape design. After the match between the material and the shape was determined by the method of perceptual image, it was necessary to design the combination and transformation of the shape. The transformation of the shape was based on fractal theory, which is a theory for studying non-obvious geometry and a method specifically used to describe the geometric shape of natural forms. It is commonly used in natural forms that cannot be expressed with regular length, area, volume, and geometric figures, such as the surface of mountains, coastlines and branches. By using fractal theory, the unity of the element body and the shape, and a combination of order and repeatability can be reflected. Fractal is made of parts that are similar to the whole to a certain extent and have self-similar characteristics. If a fractal object is enlarged, the element body in the same form will appear repeatedly. The self-similarity of fractal is reflected in the consistency of the whole and the part, which is implied in the emphasis of unity and order in the design. The form of natural materials also has the characteristics of complexity, while complexity is a key variable for understanding the aesthetic experience in design. In essence, it's difficult to design the complexity of natural forms.

Since Mandelbrot discovered the connection between fractal and nature in 1970s, designers and mathematicians have been exploring the use of fractals to express and imitate the complexity of nature, highlighting the novel beauty of complexity. Crop stalk is a natural form, with the characteristics of self-similarity and complexity. The use of a single plant of stalk in the design of element body is able to present the complexity of furniture products with lines. The outer frame of the furniture adopts the rhombus, cylinder, square and ring with the highest recognition in the sample experiment. The self-similarity of the element body is used to design the modular product family, and the four samples are designed with fractal theory respectively, as shown in Figure 12. Referring to the above research foundation, four samples were designed into different shapes, as shown in Figure 10. Through attitude scaling analysis of four samples, it is found that the most recognized one was sample 3, which combined sorghum stalks with cube shape, with the emotional factor extracted being "calm and stable", followed by sample 1, which is in diamond shape, and then sample 1 and sample 3 are combined together and designed into a set of furniture products with a unified style as a whole (as shown in Figure 13), which is more in line with the "overall harmony" in the perceptual factor.

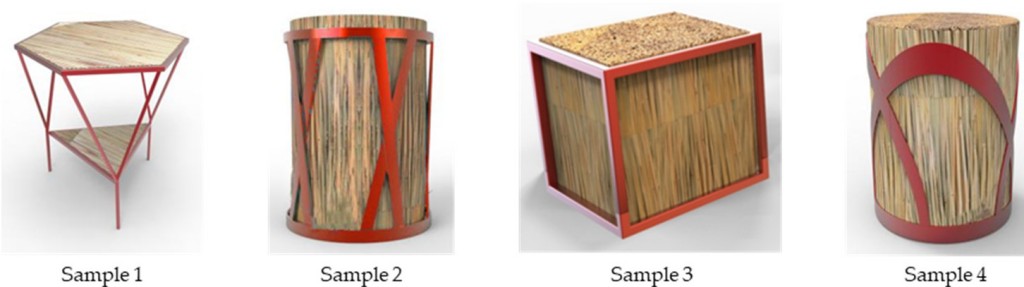

**Figure 12.** Stalk furniture shape design drawing (virtual design by author).

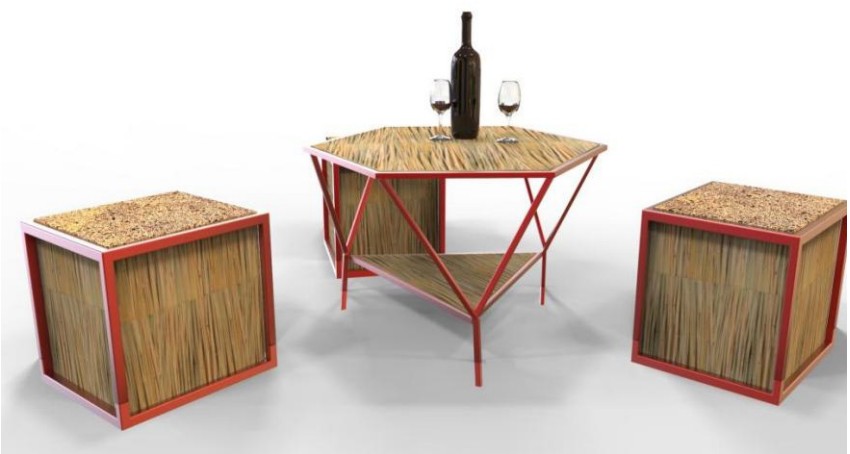

**Figure 13.** Combination design of stalk furniture (virtual design by author).

(4) Color relationship. The color of sorghum stalks remains natural and unchanged, and the product can be enriched by changing the color of the combined material. The combined metal materials are sprayed with various colors of paint on the surface rather than a single color, which is conducive to the design and production of different personalized furniture products [24].

(5) Style expression. The rural-style stalk furniture, in view of the word "animal and plant image" extracted from the perceptual factors of the previous evaluation grid diagram, make use of insects in the field as an expression element, blending the image of insects into its design to reflect its style [25]. The outer frame of the furniture is made of stainless

steel, which retains the original luster of the metal, so that it reflects the concept of "natural simplicity" and "integral harmony".

*Product Design Verification*

According to the results of the questionnaire, sample 3 is more consistent with the pastoral style image than other samples. The previous research results are optimized for design. First, on the basis of sample 3, "animal and plant images" such as insect in the field will be added as decorations; second, sample 3 and sample 1 are designed in combination, with stainless steel used as the material of the outer frame. The final design effect is shown in Figure 14.

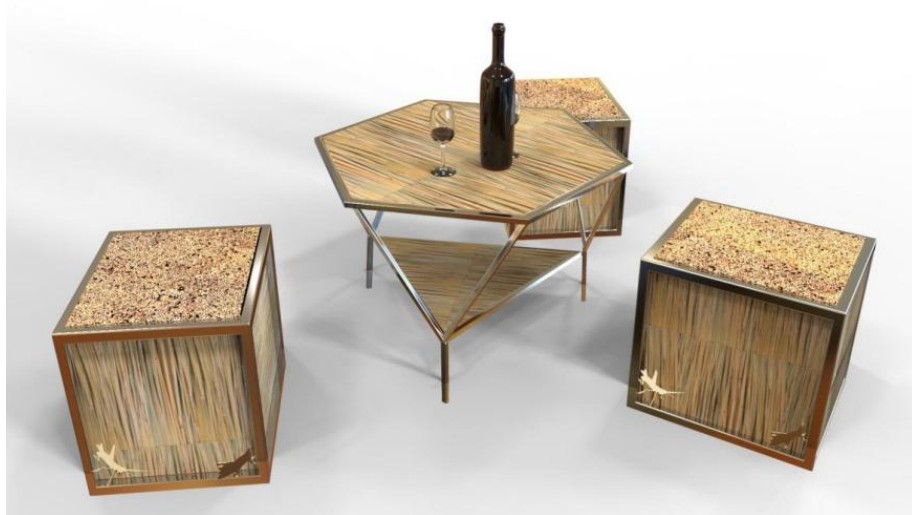

**Figure 14.** Improved rural-style stalk furniture design (the virtue design by author).

In order to verify the effect of the new design, a questionnaire was conducted again to evaluate the improvement of products. The evaluation content of the questionnaire is the scores of the 4 main charm factors of the sample 3 before the improvement and of the rural style after the improvement, using the Likert scale to score, with the highest 5 points for satisfaction, and the lowest 1 point for dissatisfaction. Through the scoring by online platform Sojump, 51 questionnaires were recalled, 46 of which were valid. The comparison of statistical results shows that the score values (Table 4 and Figure 15) of the optimized combination design is higher than that of the sample 3. Among them, the original score of the factor "overall harmony" has increased from 3.46 to 4.27, indicating the improved design of a combination is closer to the desired design objective than the sample 3. The experiment reveals that the charm factors extracted by the evaluation grid method have effectively improved the accuracy of the design and are closer to the expected target [26]. Through the binding experiment of material properties, aesthetics and perceptual imagery, the effective use of waste stalk materials has been realized to meet people's functional and spiritual needs. In the life cycle of stalk furniture, they are easy to be disassembled and recycled, and can be naturally degraded, which conforms to the concept of sustainable development, and provides a new way for the reuse of waste stalk materials.

**Table 4.** Comparison of charm factors before and after the improvement.

| Improvement | Gentle and Elegant | Wholly Harmonious | Natural and Simple | Calm and Steady | Average Value |
|---|---|---|---|---|---|
| Before improvement (Sample 3) | 2.91 | 3.46 | 3.32 | 3.41 | 3.28 |
| After improvement (combination design) | 3.12 | 4.27 | 4.11 | 4.31 | 3.95 |

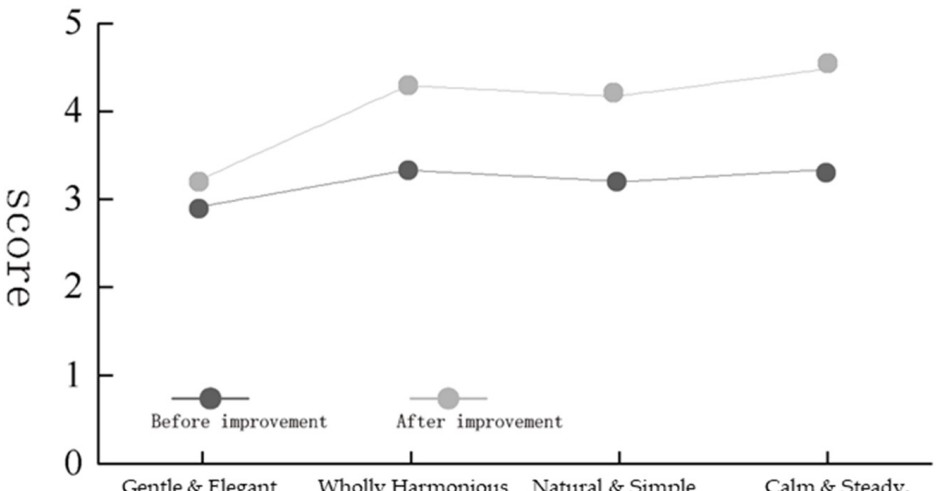

**Figure 15.** Comparison of design score of samples 3 after improvement.

## 4. Discussion

This research has explored the use of waste stalk as an element for furniture design from a design perspective. The presumption of it is the design of rural-style furniture, through the use of the evaluation grid method in Miryoku engineering, with the user's emotional needs analyzed. In the process of exploring the emotional cognition of rural-style stalk furniture, the research establishes a comprehensive product evaluation chart [27], mainly through the upper, median, and lower factors in the charm factors; then, the emotional needs of the respondents can be obtained by comparing the implemented survey questionnaires, and further, the emotional identification of the selected samples by them can be achieved. Therefore, the designer can carry out design practice activities based on the evaluation of respondents. This research has discovered the mapping relationship between representative samples and user emotional components. The design and application of waste stalks can start with the positioning of style design of furniture, combined with the diagrams of user evaluation; designers can accurately design products that users like, and meanwhile, understand the characteristics and new value of waste stalk materials.

This paper mainly conducts design research from the dimensions of stalk furniture's shapes, aesthetics, and perceptual cognition, based on the physical properties of stalk materials. It does not discuss the deep processing of stalk materials and their transformation under chemical reactions. The evaluation grid method used in this research is sometimes affected by other factors in the in-depth interviews for emotional needs exploration, which has influence on the accuracy of judgment and ultimately affects the designer's design practice.

In view of the remaining shortcomings in the paper, the following three aspects will be studied in the future:

(1) To expand the research field of stalk materials, it is better to study its material characteristics, explore the possibility of using it as a design material from the perspective of deep processing, combining perceptual knowledge with rational thinking, and better recycle waste stalk materials by turning it into the wealth in daily life. Eventually, it could reduce air pollution caused by combustion and achieve sustainable development.

(2) In the perceptual evaluation process, it must avoid variations under the influence of the in-depth interviews. After collecting the products' charm factors, the study is able to conduct a more accurate calculation with quantification-I theory [28]. The theory involves the study of the relationship between a set of qualitative variables x (argument) and a set of quantitative variables y (the dependent variable). With multiple regression analysis, it could build a mathematical model between them where the dependent variable can be predicted. The purpose of using quantification-I is to obtain the influence weight of product charm and predict the variability of

external benchmark data and events [29]. Then, it assists to identify the charm factors of the product and provides a basis for the design. Via quantification-I algorithm, each "lower specific evaluation item" is regarded as "independent variable (x)", with 1 representing the sample having its characteristics and 0 representing the sample that does not have its own characteristics. According to these differences, then, the study set the rating score of charm obtained in the previous step as "dependent variable (y)", and created the relevance of design elements and charm level by the relational expression of regression analysis analyzing the impact of "lower specific evaluation item" on the design attractiveness of product in order to establish a more refined prediction model [30].

(3) This research combines the style design of product with the user's emotional cognition. Hence, the most important factor for designers is to consider the effectiveness of design symbol extraction. From the results of the analysis, the designer's design cognition is based on the understanding of the user's emotional cognition, which is related to the effectiveness of design symbol extraction. Therefore, it is necessary to establish an in-depth communication model between the two, including the interviewee's interpretation of this perceptual knowledge as a feature of design symbols, which will improve the accuracy and reference value of the prediction results [31].

## 5. Conclusions

This study has explained the entire process and method of designing the reuse of waste stalk on the presumption of reducing damage to the ecological environment. The application of stalk materials in the design of furniture products through practice provides a new way for the effective use of waste stalk materials. By applying the evaluation grid in Miryoku engineering as the theoretical framework, and combining qualitative research with quantitative research, this study has extracted the user's overall feelings about the rural-style stalk furniture. In the method setting, the reference photos provided to respondents in this study may be too subjective, resulting in the weak objectivity of the selected photos. However, the research steps developed through this study still can obtain users' evaluations of the simple pastoral style of stalk furniture products by implementing questionnaire surveys. Consequently, it could help designers screen out materials and shapes more accurately that matches the imagery expression of the furniture appropriately [32]. The verified design shows that the perceptual image of stalk materials meets the emotional needs of users for furniture, and at the same time, it proves the feasibility of using waste stalk materials in the style design of furniture products.

The results show that stalk materials can be used as substrates and surface materials in furniture design, providing good effects and unique aesthetics. Stalk is a supplement to the visual effects of industrial materials. After the completion of a cycle of the stalk furniture products, the waste stalk can be replaced and biodegradable, so that they can be sustainably recycled, without polluting the natural environment. The evaluation grid method proposed in this research not only effectively establishes the connection between subjective feelings and design features, it also helps to accurately apply stalk materials in different furniture design, conveying the green concept of applying natural materials in furniture design ideas. The research results can be used as a reference strategy for the ecological treatment of waste stalk. These relevant experiences and methods could create a reasonable mechanism to be applied to the innovative design of stalk furniture products [33]. The direction of future research could further emphasize building a universal model of detection mechanism, and in particular, on the possibility of secondary or tertiary reuse or environmental recovery. As a result, this model could provide relevant stakeholders with judgment and evaluation on the reuse of crop waste, so as to meet the long-term goal of environmental and social sustainable development for human beings.

**Author Contributions:** Conceptualization, H.L. and K.-H.W.; methodology, H.L. and K.-H.W.; software, H.L.; validation, H.L. and K.-H.W.; formal analysis, H.L. and K.-H.W.; investigation, H.L.; resources, H.L.; data curation, H.L.; writing—original draft preparation, H.L.; writing—review and editing, H.L. and K.-H.W.; design, H.L. All authors have read and agreed to the published version of the manuscript.

**Funding:** Intramural Mentorship Research Project (2021HSDS21), sponsored by Guangzhou Huashang College.

**Institutional Review Board Statement:** Not applicable.

**Data Availability Statement:** Not available.

**Acknowledgments:** The authors are extremely grateful for the anonymous' valuable comments on improving the quality of this article.

**Conflicts of Interest:** The authors declare no conflict of interest.

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
