# Peer review of "Research on Design of Stalk Furniture Based on the Concept and Application of Miryoku Engineering Theory"

_sustainability, doi:10.3390/su132413652_

Round 1

Reviewer 1 Report

  • This article through the content of Miryoku Engineering research is to quantify human emotions that are difficult to analyze and capture where its conclusions could produce effective and quantitative support for design style analysis. However, the description of sample collection is not objective, and its operation process must be clearly stated to further understand whether it is representative.
  • (3) Interviewees: the selection criteria and the reasons for the allocation of the 8 interviewees mentioned? Please further explain
  • Please further explain how the process of establishing the evaluation grid diagram (EGM) , how to converge and the degree of objectivity? Now only shown in Figures 6-8, it is difficult to evaluate the objectivity of the process.
  • This article through the online scoring platform Sojump, a total of 113 questionnaires were retrieved, 27 invalid questionnaires were eliminated, and the remaining 86 questionnaire results were counted, and the average and overall average values of the 4 samples in each rural style charm factor. The sample number of the questionnaire is insufficient and the validity of the answers cannot be grasped. It is necessary to further explain the remedy method?
  • Referring to the overall evaluation grid of the rural style stalk furniture, charm factors and the specific performance features of the rural style are sorted out, meanwhile, the design elements are extracted; What are the objective reasons for the proposed design strategy? Please explain

  • This article points out in the conclusion: Although in the method setting, the reference photos provided to respondents in this study may be too subjective, resulting in the weak objectivity 516 of the selected photos. However, the research steps developed through this study still can obtain users' evaluations of the simple pastoral style of stalk furniture products by implementing questionnaire surveys. However, its objectivity is still questioned, please add an explanation.

Reviewer 2 Report

This is an interesting study that aimed to discuss the reasonable and effective use of waste stalks, and to explain the excellent characteristics of stalks from the perspective of green and ecological sustainable design, and the application of stalks in furniture design. Here are my comments:

  1. I believe that the aim of the study displayed in the abstract should be consistent with the aim of the study with the introduction (towards the end just before Materials and Methods), and the achievement of the aim should be explicitly stated in the conclusion. Please make sure all the statements are consistent.
  2. The sample size of the interview and the questionnaire should also be stated in the abstract.
  3. The sampling method for the interview study should be stated in the abstract and method sections. For instance, I believe your team interviewed various professionals from specific backgrounds. So, it is likely you are using purposive sampling (you may suggest otherwise if it is not).
  4. Were there any comparisons between Miryoku Engineering with other concepts or frameworks or design approaches? For instance, how does this method fair against design thinking, design for usability, TRIZ etc? Perhaps a short paragraph about this can be included under the method section (with citations and justifications).

Thank you and all the best.
